# Cost-Effective Next Generation Sequencing-Based STR Typing with Improved Analysis of Minor, Degraded and Inhibitor-Containing DNA Samples

**DOI:** 10.3390/ijms24043382

**Published:** 2023-02-08

**Authors:** Sara-Sophie Poethe, Julia Holtel, Jan-Philip Biermann, Trine Riemer, Melanie Grabmüller, Burkhard Madea, Ralf Thiele, Richard Jäger

**Affiliations:** 1Institute for Functional Gene Analytics, Bonn-Rhein-Sieg University of Applied Sciences, Grantham Allee 20, 53757 Sankt Augustin, Germany; 2Department of Natural Sciences, Bonn-Rhein-Sieg University of Applied Sciences, von-Liebig Str. 20, 53359 Rheinbach, Germany; 3Department of Pediatrics and Adolescent Medicine, Experimental Neonatology, Center for Biochemistry, Medical Faculty and University Hospital Cologne, University of Cologne, Joseph-Stelzmann-Str. 52, 50931 Cologne, Germany; 4Institute of Legal Medicine, University of Bonn, Stiftsplatz 12, 53111 Bonn, Germany; 5Computer Science Department, Hochschule Bonn-Rhein-Sieg, University of Applied Sciences, Grantham Allee 20, 53757 Sankt Augustin, Germany; 6Institute of Safety and Security Research, Hochschule Bonn-Rhein-Sieg, University of Applied Sciences, Grantham Allee 20, 53757 Sankt Augustin, Germany

**Keywords:** forensic, short tandem repeat, DNA profile, high-throughput sequencing, next generation sequencing, massive parallel sequencing, bioinformatics, degraded DNA, PCR inhibitors

## Abstract

Forensic DNA profiles are established by multiplex PCR amplification of a set of highly variable short tandem repeat (STR) loci followed by capillary electrophoresis (CE) as a means to assign alleles to PCR products of differential length. Recently, CE analysis of STR amplicons has been supplemented by high-throughput next generation sequencing (NGS) techniques that are able to detect isoalleles bearing sequence polymorphisms and allow for an improved analysis of degraded DNA. Several such assays have been commercialised and validated for forensic applications. However, these systems are cost-effective only when applied to high numbers of samples. We report here an alternative, cost-efficient shallow-sequence output NGS assay called maSTR assay that, in conjunction with a dedicated bioinformatics pipeline called SNiPSTR, can be implemented with standard NGS instrumentation. In a back-to-back comparison with a CE-based, commercial forensic STR kit, we find that for samples with low DNA content, with mixed DNA from different individuals, or containing PCR inhibitors, the maSTR assay performs equally well, and with degraded DNA is superior to CE-based analysis. Thus, the maSTR assay is a simple, robust and cost-efficient NGS-based STR typing method applicable for human identification in forensic and biomedical contexts.

## 1. Introduction

Forensic DNA typing is currently based on a set of highly polymorphic short tandem repeat (STR) loci, the alleles of which differ in the number of repeat units (reviewed in [1]). To establish DNA profiles, these STR loci are amplified by multiplex PCR using primers that hybridise to the flanking regions encompassing the repeat regions. For the different alleles, this results in different lengths of respective PCR products that are analysed using capillary electrophoresis (CE); detection of PCR products is accomplished by the use of fluorophore-labelled primers in the multiplex PCR, resulting in an electropherogram, in which the loci are displayed in four or five different colour channels [2,3]. In the current German forensic system, a set of 16 STR loci is routinely analysed using commercial and validated multiplex PCR kits [4]. The fluorescent colours are assigned to the PCR amplicons in such a way that on the electropherogram within one colour channel the size ranges of individual loci do not overlap, thus allowing for unambiguous identification of the alleles of each locus.

PCR-based STR analysis faces several common challenges. First, at low numbers of DNA copies, due to stochastic sampling effects (that result from unequal copy numbers in the sample or from unequal amplification in the first PCR rounds), alleles of STR loci or complete loci may be underrepresented (causing imbalanced STR profiles) or even drop out (allele dropout, ADO; locus dropout, LDO) [5]. In current STR kits, the limit of sensitivity for detection of complete STR profiles is in the range of 100 pg human genomic DNA (which corresponds to approximately 15 diploid cells); at lower DNA amounts, stochastic effects will ensue [6,7,8,9]. A second problem may arise from the presence of so-called PCR inhibitors—compounds derived from the traces, such as heme, humic acid, melanin or fabric dyes that may be present in the DNA extract and may impair PCR by various mechanisms [10,11]. Modern STR kits are rendered robust against PCR inhibitors by non-disclosed supplements in the PCR buffer; bovine serum albumin (BSA) has been published as one such suitable supplement [12]. Furthermore, due to environmental influences (such as heat, acidic pH or the presence of DNases), frequently the DNA from forensic traces is partially degraded, thus exhibiting DNA damage, strand breaks and deletions at random positions (see [13] for review). Longer STR amplicons are more likely to be affected by DNA degradation and thus prone to ADOs or LDOs. Attempts to overcome the limitations imposed by DNA degradation have thus been based on designing shorter STR amplicons [14,15]. However, inevitably, several loci have to be covered by longer amplicons because in the CE method, within one colour channel the size ranges of individual loci must not overlap.

Analysis of STR fragment lengths is also possible by sequencing, and next generation sequencing (NGS) methods are suitable for amplicon sequencing of PCR multiplexes [16]. Thus far, two NGS methods have been commercialised and validated for forensic purposes: an STR typing assay based on the ion torrent principle [17], and two STR typing assays for the Illumina MiSeq platform, that is based on bridge amplification followed by sequencing by synthesis (SBS) [18,19]. These assays are able to reveal isoalleles with the same fragment lengths but differences in sequence, by this means increasing discriminatory power [20]. In terms of DNA degradation, NGS methods offer the advantage over CE-based analysis of allowing for the design of overlapping allele size ranges, because identification of loci is based on the sequence, not length of PCR fragments. By this means, for all loci, shorter amplicons are possible, improving the analysis of degraded DNA [21].

Commercial NGS systems have the disadvantage of being designed for high throughput sequencing, sometimes using specifically adapted instrumentation with dedicated software for allele identification, making analysis less flexible and expensive, and not cost-effective for low throughput analysis. In this paper we describe a cost-effective low sequence-output NGS assay (called maSTR NGS, for mini-amplicon STR NGS) for the Illumina MiSeq platform allowing for DNA profiling of the 16 STR markers (plus the amelogenin sex marker) tested in Germany. Small amplicon sizes were chosen to improve the analysis of degraded DNA samples. Furthermore, the maSTR assay has been adapted to the low throughput MiSeq Reagent Nano Kit v2 and was rendered robust against common PCR inhibitors. Validation of the maSTR assay in comparison to a commercial CE-based STR kit revealed comparable sensitivities and improved analysis of degraded DNA samples.

## 2. Results

### 2.1. Characteristics of the maSTR Assay

The maSTR assay is a targeted NGS approach that analyses the 16 forensic STR loci (plus the sex marker amelogenin) tested in Germany following their amplification in a multiplex PCR. Primers binding to the flanking regions of the loci have been designed to generate short amplicons including the repeat regions and known adjacent SNPs (for primer sequences see Materials and Methods, Section 4.3). As shown in Table 1, for most STR loci, the amplicons are considerably shorter than those of the CE-based PowerPlex ESX17 kit.

To perform low-cost NGS-based forensic DNA typing, the MiSeq Reagent Nano Kit V2 was used with a sequence output of 0.5 Gb corresponding to 1 million clusters and to 2 million paired-end reads. The total sequence yield obtained for the sequence run was 0.5 Gb with 0.4 Gb having a quality score equal or higher than Q30, which was in line with the expected specifications for the MiSeq system and for the type of the sequencing kit used [22]. The cluster density was 749 ± 3 k/mm^2^, indicating an underclustering issue [23]. Moreover, pronounced differences in the proportion of nucleotides within and between sequencing cycles indicated a low sequence diversity that is typical for amplicon-based libraries. The raw sequence data obtained for a maSTR NGS run were analysed with a bioinformatic pipeline, called SNiPSTR, specifically developed for this study. SNiPSTR assigns the reads of the 16 different STR loci and amelogenin to the respective alleles, as well as identifies stutters and other PCR artefacts. The results are summarised in form of an Excel sheet listing the sequences of all reads of the different loci. Additionally, bar charts, which plot the number of reads against the alleles of each of the markers, are generated. Examples of bar charts are shown in Appendix A.

As illustrated with two examples by Table 2, the sequence information can be used to discriminate between the sources of alleles. As shown for D21S11, one can distinguish between allele 29 from contributor A5 and the stutter caused by allele 30 from contributor B5. For SE33, the sequence information allows for discriminating the isoalleles of the two contributors.

### 2.2. Study Design

To compare the performance of the maSTR assay to STR typing by CE using the commercial PowerPlex ESX17 kit [6] (referred to as CE typing hereafter) as a benchmarking standard, four separate studies were designed using simulated forensic samples with commercially available, anonymous human DNA. In each study causative parameters linked to forensic performance issues were varied and analysed for their impact on the STR typing performance.

As such, (i) sensitivity was studied using samples with different amounts of input DNA from one individual donor, DNA A5; (ii) mixtures of DNA originating from more than one individual were studied using DNA from two human individuals (DNA A5 and DNA B5) mixed in different proportions; (iii) degradation issues typical of forensic applications were studied using HeLa cell DNA treated with different amounts of DNase I; (iv) the effects of PCR inhibitors on STR typing success were studied by adding to the DNA samples known forensic PCR inhibitors in varied concentrations. Representative examples of maSTR assay results of all experiments are shown in Appendix A. Electropherograms of the DNA samples analysed are shown in Appendix A.

These simulated forensic DNA samples were then subjected to the respective PCR-amplification workflows as required for maSTR assay or CE typing in order to derive from the same input samples comparative back-to-back STR typing of the 16 German forensic STR loci plus the amelogenin sex marker.

STR typing performance was assessed in terms of allele recovery, inter-locus, and intra-locus balances, where the allele recovery represents the percentage of correctly called alleles, and the inter-locus and intra-locus balances assess whether generated PCR products are homogeneously represented either within one heterozygous locus (intra-locus balance), or between the loci (inter-locus balance).

### 2.3. Sensitivity Study

For the sensitivity study, human genomic DNA samples with different DNA input amounts ranging from 1 ng down to 31.25 pg were analysed by the two methods. The maSTR assay was tested with three replicates for 500 pg, 62.5 pg and 31.25 pg input DNA amount and with two replicates for the remaining input DNA amounts. For the two lowest DNA amounts, an additional replicate with BSA included in the PCR buffer was analysed. For CE typing, one replicate was analysed for all DNA amounts. Figure 1a shows allele recoveries calculated for each DNA amount and method tested. Allele recoveries of 100% (no allele loss) in all three replicates were achieved by the maSTR assay at all DNA amounts, except at 31.25 pg where one of the replicates displayed one ADO. The CE method displayed ADOs at DNA amounts of 62.5 pg and less. Please note that for the two lower input DNA concentrations the maSTR assay was in addition tested with BSA included in the PCR buffer.

The inter-locus balance was assessed and expressed as the relative standard deviations (RSD) of the read numbers (or the relative fluorescence intensity for CE typing) between the loci (Figure 1b). Lower RSD values are, thus, indicative of more balanced profiles. Generally, the RSDs of the samples analysed with the maSTR assay were higher than with CE, probably due to several amplification steps involved in the NGS method.

To assess the intra-locus balance, peak ratios of the eleven heterozygous loci of DNA A5 were determined (Figure 1c,d). For most loci, only moderate changes in the intra-locus balance were observed for the maSTR assay at DNA amounts from 1 ng to 125 pg, and peak ratios were comparable with those of CE-based analysis. At DNA amounts lower than 250 pg, for some loci high variations were seen between the replicates. For DNA amounts less than 125 pg, the loci D1S1656 and D2S441 showed low peak ratios when analysed by maSTR assay. Please note that, apart from 500 pg DNA amounts, for CE analysis only one replicate was analysed. The ADOs observed in four STR loci thus resulted in peak ratios of zero at the two lowest DNA amounts. Comparable intra-locus balances between maSTR assay and CE analysis were also obtained when analysing 500 pg DNA from HeLa cells where three additional loci (D8S1779, D16S539, and D21S11) are heterozygous (see Appendix A).

### 2.4. Degradation Study

For the degradation study, DNA from HeLa cells was artificially degraded with different amounts of DNase (Figure 2a). The maSTR assay was tested with two replicates. For the highest DNase concentration, an additional replicate was analysed, as well as one sample with BSA included in the PCR buffer. For CE typing three replicates were analysed.

Complete profiles were obtained for the maSTR assay-typed samples up to 1.19 mU µL^−1^ of DNase. For 2.38 mU µL^−1^ DNase, with the maSTR assay, allele recovery of 83.3% was achieved with the locus D12S391 dropping out and with single ADOs at the loci D16S539, D19S433, and D2S441. Complete profiles were obtained by CE with up to 0.6 mU µL^−1^ of DNase. However, at higher DNase concentrations, electropherograms showed decreasing peak heights with increasing amplicon size and drop-outs for the largest amplicons.

Accordingly, in terms of both intra- and inter-locus balance the maSTR assay performed superior compared to CE typing (Figure 2b,c). The generally low peak ratios obtained with both methods (Figure 2b) resulted from several loci that are disbalanced in HeLa DNA, probably due to the aneuploidy and other karyotypic characteristics of this cancer cell line [24]. The RSDs remained similar up to DNase concentrations of 1.19 mU µL^−1^ DNase. The apparent increase of the RSD in CE typing for DNA treated with 2.38 mU µL^−1^ DNase is a mathematical consequence of the many ADOs observed at this DNase concentration.

### 2.5. Mixture Studies

To test the ability of the maSTR assay to determine the STR profile of two contributors in DNA mixtures, samples with DNA of two contributors, A5 and B5, were analysed. DNA of A5 and B5 were mixed in ratios ranging from 50:50 to 98:2 and total input DNA amount was kept constant at 500 pg. The STR profiles of DNA A5 and B5 determined with the 500 pg DNA samples are listed in Appendix A. For the maSTR assay, two replicates were analysed for all ratios except for 98:2 where three replicates were analysed. For CE typing, one replicate was analysed for all ratios. For all methods, complete profiles of the major contributor DNA A5 were achieved for all samples. The allele recoveries for the minor contributor B5 obtained with both methods for the different mixture samples are summarised in Figure 3a. For the 50:50 and 75:25 mixtures, complete STR profiles were achieved with both methods. For the 90:10 and 95:5 mixtures, the allele recoveries decreased much stronger for the maSTR assay than for CE, indicating that the maSTR assay is less suitable for the analysis of DNA mixtures with low amounts of the minor contributor’s DNA. The concordances for the 98:2 mixture sample were similar for both methods. However, most of the called alleles of this sample were those shared with contributor A5 and thus could not be attributed to the minor contributor.

With decreasing amounts of DNA B5 in the mixture, the intensities of the alleles of DNA B5 became much lower, which was reflected by the continuous decrease of the peak ratios (Figure 3b). The low signals or low read numbers of alleles of contributor B5 for the 98:2 mixture can be seen in Appendix A. Some alleles of contributor B5, which were at an n-1 position of an allele of DNA A5, were not called by CE and maSTR assays since they fell below the stutter thresholds. The profile of B5 contained six heterozygous loci that did not share alleles with A5 and thus could be evaluated in terms of intra-locus balances. As shown in Figure 3c,d, the intra-locus balances of these loci decreased with decreasing proportion of B5 in the mixture for maSTR assay and CE. Peak ratios could not be calculated in cases of allele drop-outs. In 90:10 mixtures, this was the case for one locus in both types of analysis. In 95:5 mixtures, allele drop-outs occurred at three loci in maSTR analysis and at four loci in CE analysis, and at a mixture ratio of 98:2, all loci displayed allele drop-outs in both types of analysis.

### 2.6. Inhibitor Studies

#### 2.6.1. Hematin

Complete profiles were obtained for the 30 µM hematin sample typed by the maSTR assay (Figure 4a). ADOs at the locus SE33 occurred for the 60 µM hematin sample causing the inter- and intra-locus balances to decrease (intra- and inter-locus balances for all inhibitor experiments are shown in Appendix A), indicating the initial inhibitory effects of hematin on the maSTR assay. For the 120 µM and 240 µM hematin samples, no alleles were called with the maSTR assay.

As a possible technique to overcome PCR inhibition, two modifications of the maSTR assay protocol were tested. First, the PCR master mix of the PowerPlex ESX17 kit (hereafter referred to as PowerPlex MM) was used for setting up the maSTR multiplex PCR. In a second experiment, the multiplex PCR buffer of the maSTR assay was supplemented with 0.6 µM BSA. The complete inhibition by 120 µM hematin was overcome for samples prepared with the PowerPlex MM, which led to an average allele recovery of 92.6%. Moreover, the 120 µM hematin + BSA samples yielded complete profiles as well as intra- and inter-locus balances comparable with the no inhibitor sample. As shown in Figure 1 and Figure 2, BSA had only minor effects on sensitivities with intact and degraded DNA in the absence of PCR inhibitors.

For all hematin concentrations analysed with CE, complete profiles were obtained. These results indicate that these hematin concentrations have no noticeable effect on STR typing by CE, and supplementing the PCR buffer with BSA is a simple means to render the maSTR assay robust against hematin-mediated PCR inhibition.

#### 2.6.2. Humic Acid

DNA samples supplemented with 50–400 µM humic acid were analysed by the maSTR assay and CE typing (Figure 4b). For the 50 µM humic acid sample analysed by the maSTR assay, locus SE33 dropped out for both runs and the intra-locus balance decreased notably compared to the no inhibitor control. The allele recovery and the intra- and inter-locus balances further decreased for a humic acid concentration of 100 µM or higher. Like for hematin, we tested the PowerPlex MM and the addition of BSA, and for both modifications the inhibition was partially overcome, and allele recoveries of 88.9 and 74.1% were obtained with 200 µM humic acid, respectively. Complete profiles were obtained from all humic acid samples analysed by CE, and their intra- and inter-locus balances remained in a range similar to the no inhibitor control (Appendix A).

#### 2.6.3. Melanin

The results obtained for melanin concentrations of 25–200 µM are summarised in Figure 4c. With CE, complete profiles were achieved for all melanin concentrations tested. For the 25 µM melanin sample analysed with the maSTR assay, locus SE33 dropped out in one of the two runs and an allele recovery of 85.2% was obtained for the 50 µM melanin sample. No signals were obtained for melanin concentrations of 100 µM or above. PCR inhibition by 100 µM melanin was partially overcome by using PowerPlex MM or supplementation with BSA. For these samples, allele recoveries of 83.4 and 77.8% were obtained, respectively. The inter-locus balance achieved by usage of PowerPlex MM was lower compared to the sample containing BSA (see Appendix A).

#### 2.6.4. Indigo

Indigo concentrations up to 1600 µM were analysed by the two STR typing assays. An allele recovery of 100% and no consistent tendency of decreasing peak heights was observed for both typing methods indicating a lack of inhibitory effect on PCR of these indigo concentrations (see Appendix A).

## 3. Discussion

In this study we have established and technically validated the maSTR assay, a shallow sequence output NGS assay, in conjunction with a newly developed bioinformatics pipeline called SNipSTR that generates allele profiles comparable to the results of classical capillary electrophoresis. In terms of sensitivity and mixture analysis, this assay was on par with the CE-based PowerPlex ESX17 kit used as a benchmarking standard, and inclusion of 0.6 µM BSA rendered the maSTR assay robust against common PCR inhibitors. Moreover, the maSTR assay performed superior when analysing degraded DNA.

The maSTR assay can be run on standard MiSeq sequencers, and the raw data are in principle open to bioinformatics pipelines for forensic STR typing, such as the web-based STRait Razor Online or toaSTR [25,26]. Such pipelines then have to be adapted for the maSTR assay and the user’s own data processing requirements. Our in-house pipeline SNipSTR was specifically developed for the maSTR assay and combines the stutter model of toaSTR and the length-based allele identification principle of a previous STRait Razor version [27]. The current version of STRait Razor [25] is able to resolve isoalleles and isoallele-specific stutters as well.

A major advantage of the maSTR assay in comparison with commercial forensic NGS assays consists of lower running costs and the usage of a low throughput flow cell which make small-scale analyses more affordable. As shown in Figure 5, for throughputs of 12 or 32 samples, costs per sample are much lower with the maSTR assay than with commercial NGS assays. The major contribution to costs per sample consists of the costs for sequencing library preparations and is independent of sample throughput. The costs for sequencing reagents, including flow cells, become more favourable the more samples are analysed in parallel. Detailed calculations are shown in the Appendix A. In the current study the maSTR assay has been validated for the nano flow cell with 32 samples run in parallel. The maSTR assay can be scaled up to 96 samples, but then requires the MiSeq v3 sequencing reagents and a standard flow cell. Even then, with 25.44 EUR, the total costs per sample will be lower than for the commercial systems run with the same throughput.

The PowerPlex ESX17 kit was chosen as a benchmarking standard because it analyses the same set of STR loci as the maSTR assay. In terms of its performance, the PowerPlex ESX17 kit is comparable to other current CE-based forensic STR kits [6,7,8,9]. These kits are highly sensitive and yield reliable results when applied to DNA from a variety of forensic trace types that may contain commonly encountered PCR inhibitors. Because in CE analysis, within one colour channel, amplicon size ranges of different loci must not overlap, some loci will inevitably be covered by longer amplicons and thus be prone to DNA degradation. Targeted sequencing approaches using NGS, in contrast, allow for overlapping size ranges of all amplicons, and thus allow for short amplicons for all loci. In a study by Kim et al. (2017), this advantage was demonstrated with an NGS assay for genotyping 17 STR markers [21] of which, however, only some were part of the expanded European system [4]. Likewise, the commercial NGS assays perform better in typing degraded DNA as compared to CE [17,28,29,30]. Consistent with these results, the maSTR assay outperformed the CE-based PowerPlex ESX17 kit as well, achieving almost complete STR profiles for strongly degraded DNA samples while only one or two alleles were called with CE-based analysis. Of note, for the same alleles, the STR amplicon sizes of the maSTR assay are even smaller than those of current commercial NGS systems [31,32]. In terms of sensitivity, the maSTR assay and the commercial NGS systems play in the same league as current CE-based STR kits [17,18,33], which may indicate a general sensitivity limit of multiplex PCR-based STR analysis.

The current commercial systems are analysing a larger set of STR markers, both autosomal and gonosomal, and thus provide additional information that, however, cannot be used in the German national DNA databases. On the other hand, the commercial systems do not cover the highly variable SE33 locus that is a core locus of the current German national DNA database and is among the STR loci with the longest alleles [34]. In the maSTR assay, SE33 is the locus yielding the longest PCR products. SE33 was the locus most sensitive to PCR inhibitors, and in general displayed lower read counts than the other loci (see Appendix A). Thus, at present, we recommend confirming SE33 genotypes by CE analysis. In our experiments, we consistently observed underclustering of the flow cell which may have impacted on performance. This underclustering may be due to the fluorometric library quantification method. This method also detects incomplete library products and thus may overestimate the amount of products capable of forming clusters. Further improvement in terms of coverage may thus be achieved using qPCR-based library quantification methods [35].

In its initial protocol, the maSTR assay proved much more sensitive towards PCR inhibitors than the CE-based assay. Allele calling with the maSTR assay was completely inhibited by hematin, humic acid and melanin concentrations that still allowed allele recoveries of more than 62% when analysed with CE. We speculated that supplements of the PCR buffer of the PowerPlex ESX17 kit might be responsible for its robustness, which could be confirmed by improved STR typing performance after replacing the maSTR PCR buffer by the PCR buffer of the PowerPlex ESX17 kit. High sensitivity towards PCR inhibitors has also been described for the commercial ForenSeq DNA signature prep kit, and sensitivity could be overcome by adding BSA to the PCR buffer [36]. This beneficial effect of BSA on PCR inhibition could be confirmed by our study for the maSTR assay as well, and we could show that BSA only marginally affected sensitivity and STR typing success of degraded DNA. Thus, BSA is included in the final maSTR assay protocol.

In terms of DNA mixtures, compared to CE analysis, NGS assays have been shown to be comparably effective in minor contributor identification [18,37]. Moreover, the additional sequence information provided by NGS can help in discriminating between isolalleles of different donors and facilitate the identification stutter products in the mixtures [38]. In this study, we have not taken advantage of the sequence information but have found the maSTR assay to be able to identify minor contributors down to 5% proportion in DNA mixtures.

An important non-forensic, biomedical application of STR analysis is chimerism analysis. STR analysis is routinely used to monitor blood cancer recurrence in patients treated with bone marrow transplantation [39]. NGS workflows are currently being implemented in the genetic histocompatibility testing of registry donors in clinical laboratories involved in identifying suitable donors and in monitoring the course of therapy. Therefore, inclusion of NGS-based STR typing for chimerism analysis appears reasonable, both in terms of throughput and cost effectiveness [40]. Chimerism analysis is similar to forensic mixture analysis, in that the recurrent cancer cells can be considered as minor contributors, whereas the blood cells reconstituted from the donor bone marrow stem cells represent the major contributor. As shown in this study, the maSTR assay is currently performing worse in mixture analysis than CE-based STR analysis. However, like in forensic mixture analysis, chimerism analysis might be improved by the inclusion of sequence information. Furthermore, the maSTR assay might be rendered more sensitive by first identifying discriminatory-informative loci (e.g., using CE-based STR analysis of patient and donor DNA) and in a second step removing non-informative loci from the primer mix.

Other biomedical applications of NGS-based STR analysis might comprise the authentication of tissue specimens in clinical laboratory testing [41,42], in particular in conjunction with molecular diagnosis based on NGS [43]. Moreover, the maSTR assay might be used for cell line authentication in biomedical research [44]. The advantage of the maSTR assay over commercial forensic STR assays would consist in the flexibility to modify the primer mix and to adapt and combine the libraries with those of other targeted NGS assays. A further advantage is the usage of a low throughput flow cell, which helps making analyses of small numbers of samples more cost-effective and significantly shortens the running time.

## 4. Materials and Methods

### 4.1. Sample Preparation for Sensitivity, Mixture, Degradation and Inhibition Studies

Serial dilutions of the DNA sample A5 from the Human Random Control DNA Panel (Sigma-Aldrich, Taufkirchen, Germany) were prepared in molecular grade water for sensitivity studies. In multiplex PCR, 1 ng, 500 pg, 250 pg, 125 pg, 62.5 pg or 31.25 pg were used as DNA inputs. Two-contributor human genomic DNA mixtures were prepared from the DNA samples A5 and B5 from the Human Random Control DNA Panel at 5 ratios (50:50, 75:25, 90:10, 95:5, and 98:2) with DNA A5 representing the major contributor. All samples were diluted to a total DNA concentration of 500 pg µL^−1^. Each of the two DNA samples that served as “contributors” were also analysed individually. For the degradation study, a series of degraded samples was prepared by mixing 50 ng µL^−1^ HeLa genomic DNA (New England Biolabs, Frankfurt, Germany), DNase I reaction buffer 1×, deoxyribonuclease I in concentrations ranging between 0.074–2.38 mU µL^−1^, and nuclease-free water. A HeLa DNA sample without DNase I was used as a negative control. The samples were incubated at 25 °C for 5 min. DNase I degradation was stopped by the addition of 1 µL 50 mM EDTA to the samples and incubation at 75 °C for 10 min. The samples were diluted to 500 pg µL^−1^ DNA for analysis. PCR inhibitors including hematin, humic acid, melanin or indigo, respectively, were added to PCR reactions containing 500 pg DNA sample A5. The following concentration ranges of the inhibitors were tested: 30–240 µM hematin, 500–400 µM humic acid, 25–200 µM melanin and 200–1600 µM indigo.

### 4.2. Capillary Electrophoretic STR Analysis

Autosomal STR loci and amelogenin were analysed using the PowerPlex ESX17 kit (Promega, Madison, WI, USA) in 5 µL volume for 30 PCR cycles. PCR amplification was carried out with a GeneAmp PCR System 9700 thermocycler (Thermo Fisher, Waltham, MA, USA). PCR products were analysed by capillary electrophoresis on an ABI Prism 310 Genetic Analyzer (ThermoFisher, Waltham, MA, USA). A volume of 1 µL product was denatured in 12 µL deionised HiDi™ formamide (ThermoFisher, Waltham, MA, USA) and 0.5 µL WEN ILS 500 (Promega, Madison, WI, USA) at 95 °C for 3 min. Denatured samples were injected at 3 kV for 3 s. Data was genotyped with GeneMapper™ v3.0 (Thermo Fisher, Waltham, WI, USA) with the peak amplitude threshold for allele calling set to 50 RFUs and applying default settings for marker-specific relative stutter ratios.

### 4.3. NGS Library Preparation and Sequencing

Library preparation for the maSTR assay was performed according to the 16S Metagenomic Sequencing Library Preparation protocol [45]. Primers used for amplification of 16 STR loci plus amelogenin are listed in Table 3. Primers are complementary to the flanking regions of the respective locus and contain adaptor sequences, as listed in Table 3. Multiplex PCR reactions were prepared using the Multiplex PCR plus kit (Qiagen, Hilden, Germany) by mixing 1 µL of the respective DNA sample, 0.1 µM of each maSTR primer, multiplex PCR Master Mix and water to a total volume of 25 µL. For some experiments, the reaction mix was supplemented with 0.6 µM BSA, or replaced by the PCR buffer of the PowerPlex ESX17 kit. The samples were run on the GeneAmp PCR System 9700 (Thermo Fisher, Waltham, WI, USA) under the following reaction conditions: 5 min at 95 °C, followed by 35 cycles of 30 s at 95 °C, 3 min at 60 °C, and 3 min at 72 °C, subsequent 10 min at 68 °C. PCR clean-up, index PCR, and PCR clean-up 2 were performed essentially as described in the 16S Metagenomic Sequencing Library Preparation protocol. Libraries were quantitated on a Quantus fluorometer (Promega, Fitchburg, WI, USA). The library was sequenced on the MiSeq sequencer (Illumina Inc., Berlin, Germany) using the MiSeq Reagent Nano Kit v2 (Illumina Inc., Eindhoven, The Netherlands). Quality metrics of generated sequence data was assessed by the Sequence Analysis Viewer (Illumina Inc., Eindhoven, The Netherlands).

### 4.4. Data Analysis

#### 4.4.1. Bioinformatic Pipeline

The raw sequence data obtained for a maSTR NGS run were analysed with a bioinformatic pipeline, called SNiPSTR, developed by the IT department of the Hochschule Bonn-Rhein-Sieg. The pipeline uses Cutadapt [49] and Trimmomatic [50] for adapter and quality trimming, respectively. Paired-end reads are then merged with fastq-join from the ea-utils package [51]. SNiPSTR itself is based on STRaitRazor v2 [27] and is able to generate allele profiles that are comparable to the results of classical capillary electrophoresis. In addition, SNiPSTR uses sequence information to identify allelic variants and local haplotypes. SNiPSTR works directly on fastq-files and thus has minimal preprocessing requirements.

SNiPSTR assigns a read to a known STR if it matches a pair of short oligonucleotides (recognition elements, RE) upstream and downstream of the repetitive region. The length of the sequence between the RE is then determined and converted into an allele length. The conversion takes into account the length of the motif and non-repetitive sites between the RE and the STR. The resulting allele profiles do not yet contain sequence information and are comparable to the results of a CE.

The reads are separated into a repetitive (STR) and two non-repetitive parts (flanking regions) based on the known positions of the RE. Within the STR, the motifs of the locus are identified and summarised in the common repeat notation, i.e., the motif in square brackets with the number of repetitions as index, e.g., [AATG]_5_. At this step, the sequence information is taken into account to identify isoalleles. The two flanking regions are aligned to a reference genome using the Smith–Waterman algorithm to identify possible variants. The combination of STR allele length and SNV then represents a local haplotype.

All haplotypes with less than 10 reads in total are removed as noise. Afterwards, a classification into alleles, stutters and artefacts is performed. Artefacts are all haplotypes whose frequency is below a calling threshold of 2% of the locus coverage.

The stutter model is a custom implementation of the model used by toaSTR [26]. Stutters are typical artifacts in PCR amplification of STR loci. Due to replication slippage, a fraction of products will lack one or two repeat units (N − 1 or N − 2 stutter, respectively; N representing the original number of repeat units) or may have one repeat unit in excess (N + 1 stutter) [52]. The stutter model assumes that most of the possible stutters are caused by variations in the longest uninterrupted sequence (LUS, the longest consecutive portion of the same repeat unit within a compound allele) and second longest uninterrupted sequence (SLUS) [53]. For each haplotype, nine virtual stutters are generated by truncating or elongating the LUS and SLUS, to N − 1 as the most common stutter, and N − 2, N ± 1 and N + 1 as well. For each locus, a stutter threshold ST is set that corresponds to the expected N − 1 stutter ratio, that is the reads of the stutter divided by the reads of the LUS. For the N − 2 and the N + 1 stutter, this threshold is squared (ST²); for the isometric N ± 1 stutter, it is cubed (ST³).

If the sequence of any virtual stutter matches a found haplotype, the frequency of the virtual stutter is assigned to that haplotype. Identical virtual stutters from multiple sources can be assigned to a single haplotype, the frequencies are then summed. This sum represents the expected stutter (ES) of the said haplotype. Subsequently, all haplotypes with frequencies below their ES are classified as stutters. By operating with local haplotypes, SNiPSTR implicitly incorporates isoalleles in stutter classification.

SNiPSTR assigns the reads of the 16 different STR loci and amelogenin to the respective alleles, as well as identifies stutters and other PCR artefacts. The results are summarised in form of an Excel sheet listing the sequences of all reads of the different loci, as well as their classification as alleles or stutters. Additionally, bar charts, which plot the number of reads against the alleles of each of the 16 STR markers, are generated.

#### 4.4.2. Allele Recovery, Intra-Locus Balance and Inter-Locus Balance

The allele recovery was calculated for each sample by dividing the number of alleles recovered in the sample by the total number of alleles from the reference sample (multiplied by 100 to achieve the percentage). To calculate the intra-locus balance, for all heterozygous loci, the ratio of the peak heights (or of the sequence read numbers for maSTR assays) of the allele with the lower RFU value (or lower number of reads) by the peak of the allele with the higher RFU value (or higher number of reads) was calculated. If one or both alleles of a heterozygous locus were not called, the intra-locus balance for this locus was defined to be zero. To calculate the inter-locus balance, the relative standard deviation (RSD) was calculated as a measure of the balance of the peak heights (or of the sequence read numbers for the maSTR assay) between all loci. If not indicated otherwise, for all tests with the maSTR assay, two or three replicates for each DNA concentration were analysed, and results were displayed as mean values and standard deviations. For the degradation study, three replicates of the CE method were analysed. In all other tests, for the CE assays just one sample was analysed per test and DNA concentration, because results were consistent with literature data [6].

## Figures and Tables

**Figure 1 ijms-24-03382-f001:**
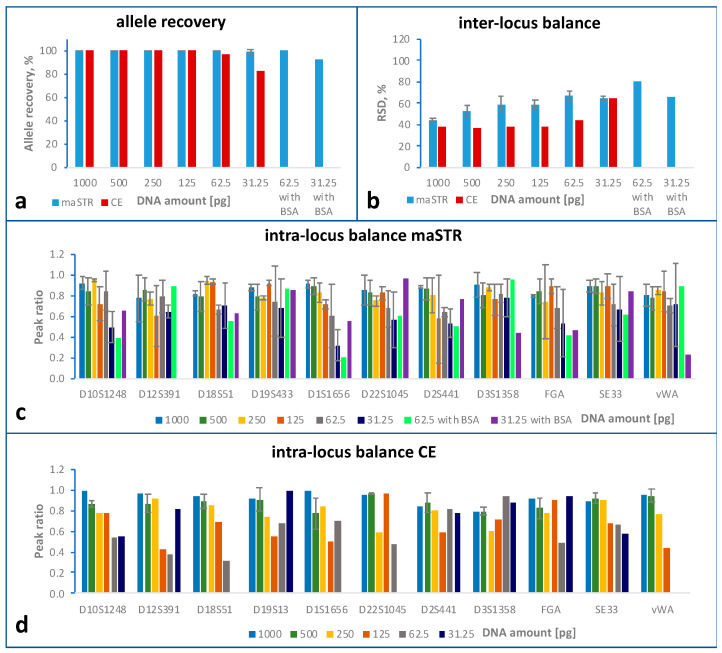
Sensitivity study comparing STR typing using maSTR assay (three or two replicates, see main text) or CE with PowerPlex ESX17 kit (one replicate). Human DNA A5 of the indicated amounts was analysed with the respective assay. For 62.5 pg and 31.25 pg DNA samples, the PCR buffer of the maSTR assay was in addition supplemented with 0.6 µM BSA (one replicate each). Error bars represent standard deviations. (**a**) Comparison of allele recoveries, which are the percentages of correctly called alleles (maSTR assay, blue bars; PowerPlex ESX17, red bars). (**b**) Inter-locus balance, expressed as the relative standard deviation (RSD) of all loci which is calculated by dividing the standard deviation of the reads (or RFUs) obtained per locus by the mean number of reads (or RFUs) per locus (maSTR assay, blue bars; PowerPlex ESX17, red bars). (**c**,**d**) Intra-locus balance of heterozygous STR loci of DNA A5 analysed with maSTR assay (**c**) or PowerPlex ESX17 (**d**), calculated as the ratio of the peak heights (or of the sequence read numbers for maSTR assays) of the allele with the lower RFU value (or lower number of reads) by the peak of the allele with the higher RFU value (or higher number of reads). The different DNA amounts are indicated by the different colours.

**Figure 2 ijms-24-03382-f002:**
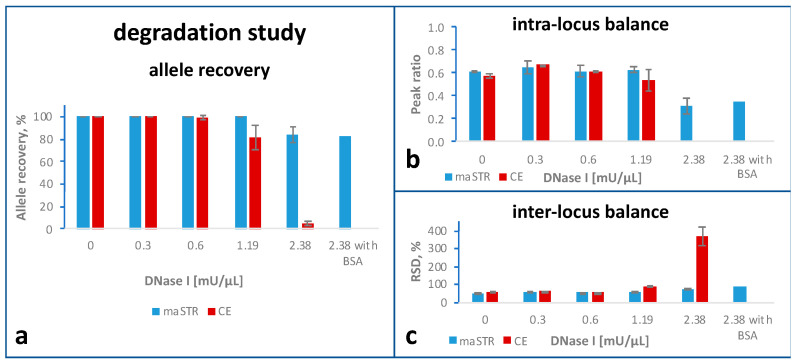
Degradation study comparing STR typing using maSTR assay (two replicates; blue bars) or CE with PowerPlex ESX17 kit (three replicates; red bars). Human DNA was incubated with the DNase I concentration indicated. For DNA samples treated with 2.38 mU µl^−1^ DNase I, the PCR buffer of the maSTR assay was in addition supplemented with 0.6 µM BSA. Error bars represent standard deviations. (**a**) Comparison of allele recoveries, which are the percentages of correctly called alleles. (**b**) Intra-locus balance, calculated as the ratio of the peak heights (or of the sequence read numbers for maSTR assays) of the allele with the lower RFU value (or lower number of reads) by the peak of the allele with the higher RFU value (or higher number of reads). (**c**) Inter-locus balance, expressed as the relative standard deviation (RSD) of the mean values (of RFUs or read numbers) of all loci.

**Figure 3 ijms-24-03382-f003:**
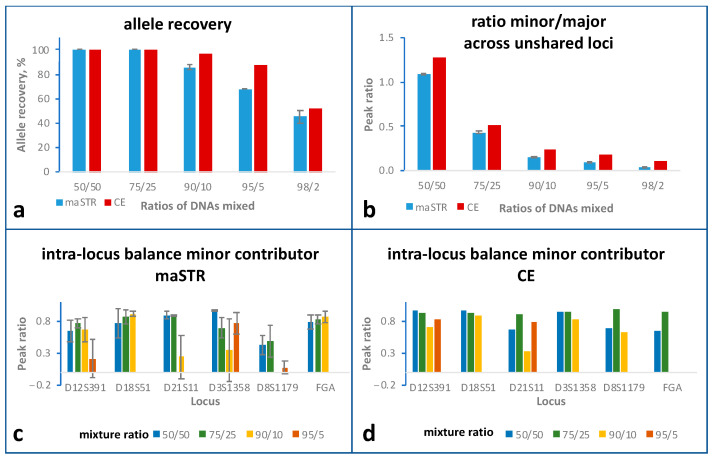
Mixture study comparing STR typing using maSTR assay (two replicates) or CE with PowerPlex ESX17 kit (one replicate). DNA from individuals A5 and B5 was mixed at the indicated ratios with a constant total DNA input of 500 pg. Error bars represent standard deviations. (**a**) Comparison of allele recoveries, which are the percentages of correctly called alleles. (**b**) Ratios of the sum of all peak heights (or of the sequence read numbers for maSTR assays) obtained for the alleles of the minor contributor to those obtained for the alleles of the major contributor. Only loci were included that did not share alleles between minor and major contributor. (**c**,**d**) Intra-locus balance of heterozygous loci of the minor contributor analysed using maSTR assay (**c**) or using CE (**d**). Only loci were included that did not share alleles with the major contributor’s profile. Please note that for the ratio of 98/2 peak ratios could not be calculated due to allele drop-outs of at least one allele.

**Figure 4 ijms-24-03382-f004:**
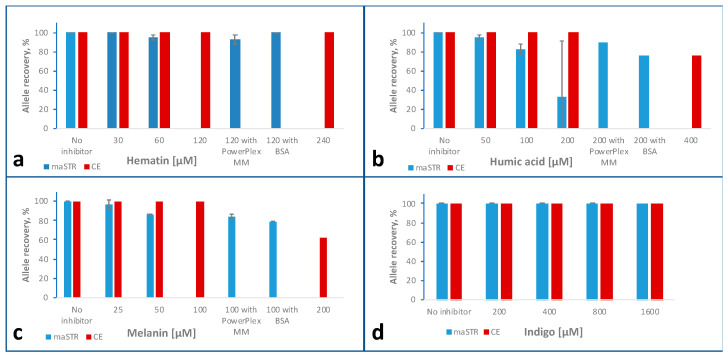
Inhibitor study comparing allele recoveries obtained with the maSTR assay (blue bars) or with CE using PowerPlex ESX17 kit (red bars). Human DNA was mixed with inhibitors of the concentrations indicated. For two samples, the PCR buffer of the maSTR assay was replaced by the reaction buffer of the PowerPlex ESX17 kit (PowerPlex MM) or supplemented with 0.6 µM BSA, respectively. Error bars represent standard deviations. (**a**) Results with hematin. Please note that with maSTR assay at 120 µM hematin, no alleles were called. (**b**) Results with humic acid. (**c**) Results with melanin. Please note that with maSTR assay at 100 µM melanin, no alleles were called. (**d**) Results with indigo.

**Figure 5 ijms-24-03382-f005:**
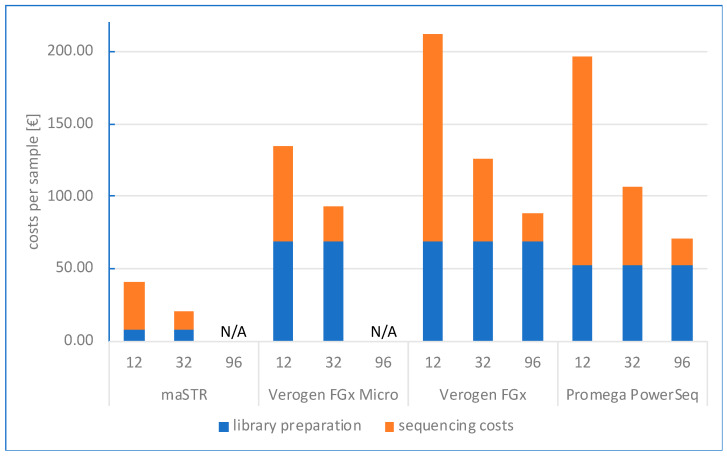
Sequencing costs, expressed as costs in EUR per sample, for the maSTR assay and three commercial, Illumina-based forensic NGS assays (Verogen’s MiSeq FGx Reagent Micro Kit and MiSeq FGx Reagent Kit, and Promega’s PowerSeq 46GY System) calculated for throughputs of 12, 32 or 96 samples, based on list prices. Please note that for maSTR and Verogen MiSeq FGx Reagent Micro Kit 96 samples are not applicable (N/A). Stacked bars represent total costs per sample, with blue and orange bars representing the proportional contribution of costs for sequencing library preparation and for sequencing reagents including flow cells, respectively.

**Table 1 ijms-24-03382-t001:** Comparison of amplicon sizes of maSTR and PowerPlex ESX17 assay.

	Allele ^1^	maSTR	ESX17
D10S1248	13	102	103
D12S391	19	165	150
D16S539	11	154	301
D18S51	18	144	330
D19S433	14	151	227
D1S1656	17	153	169
D21S11	29	174	223
D22S1045	17	123	109
D2S1338	23	146	249
D2S441	12	122	104
D3S1358	16	122	131
D8S1179	13	103	227
FGA	22	141	296
SE33	25.2	246	351
TH01	7	92	168
vWA	17	127	152
AMEL	X/Y	106/112	87/93

^1^ Alleles of the human reference genome GRCh38/hg38 (GenBank accession number GCA_000001405.15).

**Table 2 ijms-24-03382-t002:** Discrimination between stutters and alleles of the same length based on sequence.

STR Locus	DNA Donor	Classification	No. of Repeats	Sequence of Repeat Region (5′ to 3′)
D21S11	A5	allele	29	[TCTA]_4_[TCTG]_6_[TCTA]_3_TA[TCTA]_3_TCA[TCTA]_2_TCCATA[TCTA]_11_
	B5	stutter ^1^	29	[TCTA]_6_[TCTG]_5_[TCTA]_3_TA[TCTA]_3_TCA[TCTA]_2_TCCATA[TCTA]_10_
		allele	30	[TCTA]_6_[TCTG]_5_[TCTA]_3_TA[TCTA]_3_TCA[TCTA]_2_TCCATA[TCTA]_11_
SE33	A5	allele	30.2	CT[CTTT]_2_CCTTC[CTTT]_17_TT[CTTT]_13_CT[CTTT]_3_CT[CTTT]_2_
	B5	allele	30.2	CT[CTTT]_2_CCTTCCTTC[CTTT]_19_TT[CTTT]_11_CT[CTTT]_3_CT[CTTT]_1_

^1^ Stutter of allele 30.

**Table 3 ijms-24-03382-t003:** Primers used for the MaSTR assay.

Locus		Sequence 5′-3′ ^1^	Amplicon Size Range (bp) ^2^	Reference ^4^
AMEL	Fwd	CCCTGGGCTCTGTAAAGAA	106–112	[46]
	Rev	ATCAGAGCTTAAACTGGGAAGCTG		
D10S1248	Fwd	TTAATGAATTGAACAAATGAGTGAG	54–122	[15]
	Rev	CAACTCTGGTTGTATTGTCTTCAT		
D12S391	Fwd	TCAACAGGATCAATGGATGCA	149–193	tp
	Rev	ACTGTCATGAGATTTTTCAGCCT		
D16S539	Fwd	TGGGAGCAAACAAAGGCAGA	142–166	tp
	Rev	AGCATGTATCTATCATCCATCTCTG		[21]
D18S51	Fwd	CTGAGTGACAAATTGAGACCTTG	112–164	[21]
	Rev	GTTGCTACTATTTCTTTTCTTTTTCTC		
D19S443	Fwd	GCAAAAAGCTATAATTGTACCAC	99–169 ^3^	[21]
	Rev	AAAAATCTTCTCTCTTTCTTCCTCTC		
D1S1656	Fwd	GTGTTGCTCAAGGGTCAACT	125–168	[47]
	Rev	GAGAAATAGAATCACTAGGGAACC		
D21S11	Fwd	AATTCCCCAAGTGAATTGCC	156–200	[21]
	Rev	GGTAGATAGACTGGATAGATAGACGA		
D22S1045	Fwd	AGCTGCTATGGGGGCTAGAT	102–129	tp
	Rev	CGAATGTATGATTGGCAATATTTTT		[15]
D2S1338	Fwd	TGGAAACAGAAATGGCTTGG	58-162	[15]
	Rev	AGTTATTCAGTAAGTTAAAGGATTGC		
D2S441	Fwd	GGCTACAGGAATCATGAGCCA	106–138	tp
	Rev	GAGCTAAGTGGCTGTGGTGT		
D3S13358	Fwd	CAGTCCAATCTGGGTGACAG	102–134	[21]
	Rev	ATCAACAGAGGCTTGCATGT		
D8S1179	Fwd	TTTTTGTATTTCATGTGTACATTCGT	83–119	[21]
	Rev	GTAGATTATTTTCACTGTGGGGAA		
FGA	Fwd	AAATAAAATTAGGCATATTTACAAGC	121-173	[21]
	Rev	GCCAGCAAAAAAGAAAGGAA		
SE33	Fwd	GAAAGAGACAAAGAGAGTTAG	180–290	[21]
	Rev	ACATCTCCCCTACCGCTATAG		
TH01	Fwd	GATTCCCATTGGCCTGTTC	84–104	[21]
	Rev	CAGGTCACAGGGAACACAGA		
vWA	Fwd	GAATAATCAGTATGTGACTTGGATTG	103–143	[21]
	Rev	TGATAAATACATAGGATGGATGG		

^1^ The adaptor sequence for all forward primers is 5′-TCGTCGGCAGCGTCAGATGTGTATAAGAGACAG-3′; the adaptor sequence for all reverse primers is 5′-GTCTCGTGGGCTCGGAGATGTGTATAAGAGACAG-3′. ^2^ Calculated for the alleles present in the German database in STRidER [48]. ^3^ Without allele 99. ^4^ tp, this publication.

## Data Availability

The data presented in this study are available in the Appendix A of this article and upon request from the corresponding author.

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
