# Peer review of "Cost-Effective Next Generation Sequencing-Based STR Typing with Improved Analysis of Minor, Degraded and Inhibitor-Containing DNA Samples"

_ijms, 2023, doi:10.3390/ijms24043382_

Round 1

Reviewer 1 Report

Dear authors,

You spent quite some effort on providing the forensic genetics community with a small-scale assay for sequencing classical forensic STR markers. The assay performance seems robust and this definitely deserves publication. Still, a couple of questions remain open for me.

Major points:

-         From your description, it seems to me that your bioinformatics pipeline does not resolve isoalleles and isoallele-specific stutter sequences. Therefore, you lose a (if not the) main advantage of sequencing those STRs compared to only looking at length polymorphisms with CE. Please comment on this.

-          To my knowledge, it is not correct that e.g. the ForenSeq kit needs special instrumentation. It should be possible to run it on a normal MiSeq as well. What is critical is not the instrumentation, but the bioinformatics pipeline. So one could ask the question, why not only developing an open source software for the analysis of Sequencing raw data. Such a tool already exists, with STRait Razor. So what is the advantage of your pipeline over STRait Razor? This is not yet clear to me from your manuscript.

-          You claim that your approach is more cost effective than commercial panels. To illustrate this, please provide an example, calculating effective costs per sample.

-          Why did you include different numbers of replicates (sometimes two, sometimes three) in your analysis? It looks a bit arbitrary.

-          To demonstrate suitability for mixture analysis, you should demonstrate good intra-locus balances per locus and not just globally. However, the sample A5 that you selected is homozygous at 5 out of 16 loci. To assess intra-locus balance, you should analyze heterozygous alleles at every locus.

-          Why is the global intra-locus balance in the degradation blank in Figure 2b at 0.6, whereas it is at 0.8 in figure 1b?

-          The scale in figure 2c is too small, but it seems like inter-locus balance is also worse than in figure 1c.

-          The measurement of intra-locus balance with varying mixture ratios in figure 3 b seems pointless to me.

-          Please mention in the figure caption of figure 3 that allele recovery is measured by counting the alleles of A5 and B5 together, so it should in theory not go below 56% (27/48, 48 unique alleles for A5 and B5 together, 27 alleles from A5). However, it looks like your 98/2 mixture is below 50%.

-          In line 209 you state that you kept the amount of A5 DNA constant at 500pg. However, in the methods section, you state that total DNA concentration (so A5+B5) was diluted to 500pg / ul. This seems contradictory.

Minor points:

-          Too many keywords.

-          Did you compare the performance of your kit also to a commercial MPS-Kit?

-          Can you provide an explanation for the underclustering?

-          Line 163: 62.5pg should probably be 125pg?

-          Lines 193-195: the ADOs you describe where probably not the same between replicates?

-          You do not state the DNA input amount for your degradation study.

-          Line 211: should be figure 2.

-          Line 303: delete "could".

-          Line 413: RFUs or RFU, but not RFU's.

-          Maybe you want to find a different name for your assay, since MaSTR does already exist: https://softgenetics.com/products/mastr/

Best regards.

Reviewer 2 Report

- line 42: reviewed in SPACE [1]

- line 54: stiff german english, pls. rewrite

- line 55: pls explain what stochastic effects are here (regular readers will not know this)

- line 58: of human (?)

- line 72: please explain more clearly which principle of color assignments this is (regular readers will not know this). how many colours how many loci, which colours, etc.

- line 82, 86, 90, 91: pls delete german filler terms ("thus", "in particular", "furthermore")

— line 93: which common PCR inhibitors do you mean? (normal readers may not deal with our forensic case work PCR inhibitors)

- line 119: in my version, figs. S1-S35 were not visible, i could not check them

- ADO, LDO, CE → maybe the paper would be less technical if you did not use so many abbreviations throughout the text? i am not sure: i personally do not mind but maybe discuss it in your team if you wish to make the text more accessible?

- fig. 1: pls better explain RSD since many biologists have problems with standard deviation values & methods: your explanation (line 180 f.) is too stiff / hard to understand

- line 213: what exactly do you mean by calculatory consequence? this is a german english term i suppose?

- line 241: do you mean: amounts?

- line 236: total amount of input DNA → german english, pls. rewrite

- line 246: how did you define stutter in this study? relative? absolute? etc.

- line 256: as a possible technique

- line 292: fig. 

- line 299: what exactly do you mean by notably?

- line 308: technically

- line 327: outperformed: how much?

- lines 334, 335 etc.: too many "howevers"; these are german filler terms 

- line 336: please define german system: legal system? common acceptance? validation, certification, DIN ISO?

- line 338: of note (filler term)

- line 341: how much is much more?

- line 358 f.: german english, pls use shorter, clearer sentences

- line 361: in which clinical laboratories?

- line 388: 31.25 pg → 31.3 pg

- line 395: ranging

- line 411 → do you mean 3 kV?

- line 464 - 474: how is stutter defined here in more simple terms? i find the explanation difficult to understand.

- line 496/497: pls write the sentence more clear, this is german english (multi-clause schachtelsatz)

 - line 498: results were represented: do you mean shown?

- line 500/501: what do you mean by results were expected? when, how, by whom, why nor replicates necessary? pls. explain a bit.

Round 2

Reviewer 1 Report

Dear authors,

In the following, I reply to relevant points from your first revision:

"We agree that Forenseq (or Promega Powerseq) is compatible with "normal" Miseq as well and have corrected the text accordingly."

In line 28 in the abstract, you still refer to a requirement for special instrumentation. Please change.

"In the text, we keep the information generic in order to avoid legal conflicts with the vendors. We now state (line 390 ff): " While for the commercial assays, based on list prices, reagent costs range between 70 € and 85 € per sample, total costs for the maSTR assay are approximately 17 € per sample."

If you fear litigation, I am not convinced that this is the best option. List prices, without special lab discounts are public, so you don't have to fear anything in my opinion. However, by demonstrating a table with your price calculations and your assumptions (e.g., how many samples are analyzed at once) you would be fully transparent. I think your approach of just giving single numbers is more risky.

"When addressing this comment, we noticed that the description of figure 3b was in fact misleading. It actually shows the ratio of the sum of all reads obtained for the alleles of contributor B5 (minor contributor) to contributor A5 (major contributor). We have corrected this mistake. In addition we now include intra-locus balances of all heterozygous, not shared loci of B5 when analysed by maSTR (Fig 3c) or CE (Fig 3d). (See figure 3 and legend.)"

Demonstrating intra-locus balance per locus would have been especially important for figure 1 and/or figure 2! My reference to the suitability for mixture analysis was not referring to your mixture figure 3. It was a more general one, because if you have a marker in your panel that is consistently not well balanced, this one would not be suitable for mixture analysis per se, since peak height (or number of reads) interpretation would not be reliable.

You should also mention in the text, that peak height balance was not assessed for 5 out of 16 markers, due to homozygosity. This can be deduced from the profiles in the supplementary information, but it should be made clear in the text.

In addition, your revised figure 3b looks strange to me. In theory, the peak ratios minor/major for the five mixtures should be 1, 0.33, 0.11, 0.05, 0.02. Your values for 4 out 5 mixtures are consistently higher and they are even much higher for the three least balanced mixtures. This is completely counterintuitive and merits an explanation. One reason could be that you seem to ignore peaks in stutter position in your assessment. B5 has 21 unique alleles; 10 of them are in -4bp Stutter position, 3 are in +4bp stutter position to A5. Ignoring -8bp stutters, you only have 4 unbiased allele pairings to estimate minor/major ratios in CE (there may be isoalleles in maSTR): TH01, FGA (20-25), D2S441 (11-16) and D1S1656 (13-17.3). Actual contributor ratios for CE could also be estimated with probabilistic genotyping.

"In Figure 1b, the intra-locus balance is shown for the A5 DNA. Figure 2b shows HeLa DNA. Therefore, the intra-locus balances differ from each other, as profiles from HeLa DNA samples are generally less balanced (maSTR and CE), probably due to aneuploidy of the cancer cells."

It does not impair the comparison between maSTR and CE. However, it is puzzling for the reader. Therefore, if the overall bad balance in Fig. 2 is potentially due to aneuploidy, this explanation should be given in the text.

My decision to go again for a major instead of a minor revision is mainly due to: (1) The intra-locus balance that has to be better explained, with clear indications of the important limitations of the study material (if you choose to not re-assess it with samples with heterozygous alleles for all loci). (2) Figure 3b.

Best regards.

Round 3

Reviewer 1 Report

All remarks from the review have been addressed sufficiently now.